# Antenatal Betamethasone Every 12 Hours in Imminent Preterm Labour

**DOI:** 10.3390/jcm11051227

**Published:** 2022-02-24

**Authors:** Natalia Saldaña-García, María Gracia Espinosa-Fernández, Jose David Martínez-Pajares, Elías Tapia-Moreno, María Moreno-Samos, Celia Cuenca-Marín, Francisca Rius-Díaz, Tomás Sánchez-Tamayo

**Affiliations:** 1Department of Neonatology, Regional University Hospital of Málaga, 29010 Malaga, Spain; mgespinosaf@gmail.com (M.G.E.-F.); jd_martinezp@hotmail.com (J.D.M.-P.); eliastamo@hotmail.com (E.T.-M.); maria.m.samos@gmail.com (M.M.-S.); 2School of Medicine, Malaga University, 29071 Malaga, Spain; 3Department of Obstetrics and Gineocology, Regional University Hospital of Málaga, 29010 Malaga, Spain; dra.cuencamarin@gmail.com; 4Department of Preventive Medicine and Public Health, Biostatistics, School of Medicine, Malaga University, 29071 Malaga, Spain; rius@uma.es; 5Pharmacology and Pediatrics Department, Malaga University, 29071 Malaga, Spain

**Keywords:** antenatal corticosteroids, betamethasone, preterm infant, mortality, respiratory distress syndrome, neurodevelopmental

## Abstract

Background: Benefits of antenatal corticosteroids have been established for preterm infants who have received the full course. In imminent preterm labours there is no time to administer the second dose 24 h later. Objective: To determine whether the administration of two doses of betamethasone in a 12 h interval is equivalent to the effects of a full maturation. Methods: We performed a retrospective cohort study including preterm infants ≤34 weeks gestational age at birth and ≤1500 g, admitted to an NICU IIIC level in a tertiary hospital from 2015 to 2020. The population was divided into two cohorts: complete maturation (CM) (two doses of betamethasone 24 h apart), or advanced maturation (AM) (two doses of betamethasone 12 h apart). The primary outcomes were mortality or survival with severe morbidities. The presence of respiratory distress syndrome and other morbidities of prematurity were determined. These variables were analysed in the neonates under 28 weeks gestational age cohort. Neurodevelopment at 2 years was evaluated with the validated Ages and Stages Questionnaires^®^, Third Edition (ASQ^®^-3). Multiple regression analyses were performed and adjusted for confounding factors. Results: A total of 275 preterm neonates were included. Serious outcomes did not show differences between cohorts, no increased incidence of morbidity was found in AM. A lower percentage of hypotension during the first week (*p* = 0.04), a tendency towards lower maximum FiO_2_ (*p* = 0.14) and to a shorter mechanical ventilation time (*p* = 0.14) were observed for the AM cohort. Similar results were found in the subgroup of neonates under 28 weeks gestational age. There were no differences in cerebral palsy or sensory deficits at 24 months of corrected age, although the AM cohort showed a trend towards better scores on the ASQ3 scale. Conclusions: Administration of betamethasone every 12 h showed similar results to the traditional pattern with respect to mortality and severe morbidities. No deleterious neurodevelopmental effects were found at 24 months of corrected age. Earlier administration of betamethasone at 12 h after the first dose would be an alternative in imminent preterm delivery. Further studies are needed to confirm these results.

## 1. Introduction

The discovery of the effects of antenatal corticosteroid administration as a measure for assisting lung maturation in threatened preterm labour, is one of the milestones of modern medicine [1]. Since the first investigations in 1970 and 1972 by Liggins and Howie [2], several studies and reviews have been published supporting the use of antenatal corticosteroids [3,4]. They have been shown to reduce neonatal and perinatal serious outcomes (defined as death or survival with serious sequelae), neonatal respiratory distress syndrome (RDS) and they probably reduce the occurrence of severe intraventricular haemorrhage (IVH) [1,4].

Guidelines and types of corticosteroids administered have not changed much since their discovery [2]. Current international recommendations are to initiate treatment with betamethasone (12 mg every 24 h, 2 doses) or dexamethasone (6 mg every 12 h, 4 doses) [5], in any gestation at risk of preterm delivery from 24 + 0 weeks gestational age (GA) to 34 + 0 weeks GA [6,7]. Its administration should be agreed with the parents at gestational ages at the limit of viability (23 + 0 to 23 + 6). In the later preterm population, above 34 + 0 weeks GA, its use should be on an individualised basis, and there is currently much controversy in its administration, given that although prenatal corticosteroids reduce respiratory distress syndrome, more studies are needed to prove the effects on neurodevelopment in the longer term [4,7,8,9].

The benefits of prenatal corticosteroids on preterm lung maturation have been established for those pregnancies who received the full corticosteroid course and ended within 48 h to 7 days of the first dose of prenatal corticosteroid [1,4]. However, in routine clinical practice, this condition is not always met. More than 50% of pregnant women who received prenatal corticosteroids continue their pregnancy beyond day 7 from maturity or go to term [10]. It is also estimated that 15% of pregnant women who received prenatal corticosteroids finish their pregnancy before 48 h from the start of regimen, sometimes without having received the two doses of betamethasone or four doses of dexamethasone, which complete the course [11]. In these cases of imminent delivery, international guidelines recommend an administration of at least one dose of prenatal corticosteroid. [1,4,12].

Recent studies call for further research on dosing and timing of prenatal corticosteroids to individualise treatment [4,11,13].

In our centre, a tertiary hospital with a Neonatal Intensive Care Unit (NICU) level IIIC [14], maturation with prenatal corticosteroids is performed with betamethasone (50% in the acetate form and 50% in the phosphate form). In situations of imminent preterm labour, when it is not possible to wait 24 h for the administration of the second dose, it is advanced to 12 h after the first dose to complete the cycle.

The main objective of this study was to determine neonatal serious outcomes (mortality and serious sequelae) in preterm infants who received two doses of betamethasone 12 h apart compared to preterm infants who received the full standard regimen.

## 2. Materials and Methods

A retrospective cohort study was conducted from 1 January 2015 to 31 December 2020, including preterm infants ≤34 weeks, weighing ≤1500 g admitted to a tertiary hospital at a NICU IIIC level. Neonates with major malformations were excluded from the study.

Two cohorts were established with regard to the type of prenatal corticosteroid maturation received. A complete maturation cohort (CM) included neonates who received 2 doses of antenatal betamethasone 24 h apart and whose birth occurred between 48 h and 7 days after the initial dose. The advanced maturation (AM) cohort included preterm infants who received 2 doses of antenatal betamethasone 12 h apart and whose birth occurred after completion of the regimen and up to 7 days after the initial dose.

The criteria for the betamethasone every 12 h administration were: situations of imminent labour (cervical length <15 mm and regular clinical uterine dynamics despite tocolytic treatment; cervical conditions of delivery and regular clinical uterine dynamics despite tocolytic treatment; extremely advanced labour conditions where, even without uterine dynamics, labour may occur unpredictably); situations of chorioamnionitis or intrauterine growth retardation >stage III, according to Figueras classification [15] and/or with cardiotocography (CTG) recordings that are not very reassuring and situations of maternal pathology in which labour is decided to be terminated within 24 h of the first dose of betamethasone.

The following variables were recorded as demographic and clinical characteristics: maternal age, primiparity, single/multiple gestation, the presence of gestational diabetes, hypertensive stages including pre-eclampsia, eclampsia and HELLP syndrome, intrauterine growth restriction (IUGR), chorioamnionitis, and third trimester haemorrhage (placental abruption, placenta praevia, uterine rupture or vasa praevia). The following data were collected: caesarean section, gestational age (GA) in weeks, anthropometric measurements at birth, gender. The difference in hours between the first dose of corticosteroids and delivery was also determined.

In our centre, low-dose postnatal hydrocortisone was administered during the first 10 days of life in neonates under 26 weeks and between 26 + 0 and 27 + 6 weeks if they presented with chorioamnionitis or had not received prenatal corticosteroids, according to the PREMILOC protocol. The study by Baud et al. (2016) aims to increase survival and improve outcomes in bronchopulmonary dysplasia in neonates under 28 weeks gestational age [16]. Patients who received this guideline in each cohort are included in our study.

The main objective was to determine “serious outcomes”, defined as death or surviving neonates with serious sequelae, which included the following: intraventricular haemorrhage grade III–IV (IVH), periventricular leukomalacia (PVL), bronchopulmonary dysplasia (BPD), necrotising enterocolitis (NEC) stage >2, retinopathy of prematurity (ROP) requiring treatment. These variables were defined with regard to the criteria established by the Vermont Oxford working group [17,18].

As secondary outcomes, intubation required at birth was determined, Apgar score <5 at 5 min of life, the need for mechanical ventilation (MV), maximum FiO_2_ during admission and need for surfactant administration. Other variables analysed included hypotension during the first week of life requiring treatment (volume expansion or inotropic administration), patent ductus arteriosus requiring treatment, spontaneous bowel perforation and the presence of early sepsis or late sepsis [17,18].

A sub-analysis of the variables was performed for neonates under 28 weeks gestational age.

Finally, both cohorts were monitored up to 24 months corrected age. The validated Ages and Stages Questionnaires^®^, Third Edition (ASQ^®^-3) were used to assess neurodevelopmental status [19,20]. The presence of cerebral palsy was determined, defined as chronic, non-progressive impairment of motor skills, posture, balance, coordination, tone or reflexes. The presence of sensory deficits was analysed: visual (need for correcting lenses or mono/bilateral blindness) and auditory (need for hearing aids or mono/bilateral deafness).

Contingency tables and the chi–squared test were used for the comparison of qualitative variables. In the 2 × 2 tables with a low number of observations (*n* < 5), Fisher’s exact test was used. To carry out pairwise comparisons of quantitative variables, the Student’s *t*-test or the Mann–Whitney U test was used, depending on the distribution of the variables. For variables showing an association with *p* < 0.25, a multivariate model was adjusted for the following confounding factors: GA, IUGR, third trimester haemorrhage, chorioamnionitis, hypertensive stages, Apgar test score <5 at 5 min of life, gender and postnatal hydrocortisone treatment. The backward method selection was used, variables with *p* values ≥ 0.15 for the Wald statistic were removed from the model until the adjusted OR estimate was obtained. In all cases, a statistically significant difference was declared when *p* < 0.05. SPSS v25.0 (IBM Corp., Armonk, NY, USA) was used for this analysis.

This study was approved by the Provincial Ethics Committee of Málaga and by the Medical Management of the Regional University Hospital of Málaga, dated 12 November 2020.

## 3. Results

A total of 275 patients were studied, and were divided into the complete maturation (CM) cohort (*n* = 224) and the 2 doses 12 h apart (AM) cohort (*n* = 51). The clinical and demographic characteristics of the two cohorts are summarised in Table 1.

The percentage of intrauterine growth retardation and hypertensive stages were more frequent in the CM group. Gestational age at the first dose of betamethasone and gestational age at birth were similar in both cohorts. The CM cohort presented with lower weight and height at birth, with no differences in head circumference, in relation to the higher percentage of IUGR. The differences in hours from first and second dose to delivery are summarised. Gestational age at first dose of betamethasone is summarised, no differences were found.

There are no significant differences in other population characteristics, postnatal corticosteroid administration included.

The results of the overall analysis of both cohorts are summarised in Table 2. For serious outcomes, no differences were found between the two cohorts, either the components analysed individually (mortality, PVL, IVH III-IV grade, treated ROP, NEC >2nd grade and moderate-severe BPD).

With regard to the assessment of respiratory distress, no differences were found in the need for mechanical ventilation during admission, nor in the surfactant administration. Lower peak FiO_2_ and shorter mechanical ventilation time were observed in the advanced maturation (AM) cohort. When it was adjusted for confounding factors, significance was lost.

A notable finding was the lower percentage of hypotension during the first week in the AM cohort. When it was adjusted for confounding factors OR 0.5 (95%CI 0.3–0.9), it was maintained.

No cases of spontaneous intestinal perforations were reported in the AM cohort, compared to 13 cases (5.8%) in the CM cohort. There were no significant differences in cases with presence of early or late sepsis.

The variables analysed for the group under 28 weeks GA showed similar results in terms of serious outcomes and severe morbidities of prematurity (Table 3). Lower maximum FiO_2_ was found for the AM cohort. Lower hypotension during the first week of life and shorter mechanical ventilation time were found in favour of the AM group. This significance was lost in the adjusted model; however, the model decreased its goodness of fit by 12% for hypotension and 50% for mechanical ventilation time when prenatal corticosteroid therapy received, was removed from the equation.

The results of the assessment at 24 months corrected age are summarised in Table 4. No significant differences were found between the cohorts in terms of presence of cerebral palsy, visual or hearing impairment.

The results of the ASQ^®^-3 questionnaires were shown in two different ways. On the one hand, neonates who scored below the lower limit of normal on each rating scale (communication, gross-motor, fine-motor, problem-solving and social) were determined with regard to the type of maturation received. No significant differences were found between the two cohorts analysed, although the AM population had fewer patients below the lower limit of normal in all categories. On the other hand, mean scores were determined for each category analysed, with regard to the maturation pattern. In all items, the AM cohort scored better, having achieved significance for fine-motor skills and social skills. Statistical significance was lost when the model was adjusted for confounding factors, although in social skills it was close to significance (*p* = 0.06). Figure 1, box and whisker plot, shows the distribution of the population analysed with respect to the ASQ^®^-3 items, with regard to the type of maturation pattern. The AM cohort showed better scores in all categories.

## 4. Discussion

The benefits of prenatal corticosteroid administration in case of preterm birth are established for a regimen of betamethasone 12 mg every 24 h (2 doses) or dexamethasone 6 mg every 12 h (4 doses), with birth occurring from 48 h after the start of the regimen until the following 7 days [4].

This ideal situation does not always occur in clinical practice where, due to multiple factors, labour may end without having completed the prenatal corticosteroid regimen.

In our study, the effects of a full course of prenatal betamethasone were compared with a 12 h early course in situations of imminent labour.

Among the baseline characteristics of the population, a higher percentage of hypertensive stages, female gender and intrauterine growth retardation were observed in the AM cohort, so these characteristics were added as confounding factors in the multivariate analysis. With respect to the time elapsed between administration of the first and last dose of betamethasone and delivery, the differences observed are specific to the nature of each cohort.

The results shown were similar in both populations. No harm was found in terms of serious outcomes or respiratory distress syndrome, with respect to complete maturation.

Other studies have analysed the advanced 12 h schedule of antenatal betamethasone in a randomised clinical trial compared to the usual 24 h schedule. The study published by Kashanian et al. (2018) included 101 preterm infants in the 12 h group vs. 100 preterm infants in the 24 h group. Less respiratory distress and severe intraventricular haemorrhage were found in the 12 h group, but also higher mortality and necrotising enterocolitis >2nd grade. However, there was discordance in gestational ages between groups [21]. In our study, where gestational age was similar between the two cohorts, no differences were found in intraventricular haemorrhage or other morbidities. A better respiratory outcome in the 2 doses 12 h group was shown in our study, with lower maximum FiO_2_ during admission and shorter mechanical ventilation time, although both groups had equivalent intubation and surfactant administration requirements. These data were repeated in the subgroup of neonates younger than 28 weeks.

Another clinical trial published by Khandelwald et al. (2012) included 161 preterm infants in the 12 h betamethasone group vs. 77 preterm infants in the 24 h betamethasone group [22]. They divided the sample with regard to gestational age subgroups. Mortality, respiratory distress syndrome and other morbidities in preterm infants were analysed. They found similar results in both cohorts, with an increase in the incidence of necrotising enterocolitis >2nd grade. In our study, there were no cases of necrotising enterocolitis or spontaneous intestinal perforation in the 12 h group.

The haemodynamic effect of corticosteroids at foetal and postnatal level is well known [23,24,25,26,27,28,29]. The results of our investigation found greater haemodynamic stability, and a tendency to a better evolution of respiratory distress syndrome in the 12 h betamethasone group. These results were also verified for the subgroup of neonates under 28 weeks. Studies in lambs have established that the minimum prenatal corticosteroids effective dose to accelerate lung maturation is between 1–4 ng/mL. The current betamethasone regimen reaches concentrations of up to 10 ng/mL in the first hours of administration, decreasing progressively until 24 h, but maintaining the therapeutic range between 1–4 ng/mL [13]. The better respiratory and haemodynamic outcomes of preterm infants in the AM cohort may be related to the higher accumulation of prenatal corticosteroid doses. Haemodynamic evolution during the first week of life has not been previously analysed in the clinical studies referred [5,21,22]. In our centre, hydrocortisone is administered at low doses during the first ten days of life, to achieve a lower rate of bronchopulmonary dysplasia, according to the PREMILOC protocol [16]. The possible haemodynamic effect of the application of this protocol was controlled in our sample by multivariate analysis, adding it as a confounding factor. The results persisted in favour of the AM group.

The higher percentage of hypertensive stages observed in the CM cohort and therefore of neonates exposed to prenatal antihypertensive treatment (labetalol, methyldopa, hydralazine or magnesium sulphate) may influence the higher percentage of neonatal hypotension observed in this cohort [30]. However, after adjusting for confounding factors in the multivariate analysis, including hypertensive stages, significant differences persisted. To confirm our results, a sub-analysis was performed excluding hypertensive stages from the sample, and found a percentage of hypotension of 28.8% in the CM cohort (N = 132) vs. 7.7% in the AM cohort (N = 39), with a *p* = 0.007.

There are no published studies comparing the evolution of a standard complete maturation pattern with an advanced maturation pattern, with regard to monitoring at 24 months corrected age. Uncertainty about the possible deleterious effect of prenatal corticosteroids on neurodevelopment is a current problem [31,32]. No differences in sensory deficits or increased incidence of infantile cerebral palsy were found in our population. The ASQ3 questionnaire is a validated scale that provides accurate information on development in several areas: communication, gross-motor skills, fine-motor skills, problem solving and personal and social skills [33]. The scores obtained in both cohorts were similar, with a tendency towards better results in the AM group. From a neurological point of view, advancing the dose of betamethasone by 12 h did not harm our population.

Our study has several limitations to be considered. It is a retrospective cohort study, so we cannot definitively conclude that both betamethasone (every 24 h or every 12 h) treatments are similar. To confirm our results, a clinical trial of equivalence or non-inferiority should be performed.

The baseline population of both cohorts presents some differences: a higher percentage of female sex in the 12 h cohort, and a higher percentage of intrauterine growth restriction in the complete maturation cohort.

The sample size to detect a 5% difference in serious outcomes, with a 95% confidence level, should include at least 808 patients in each cohort., taking as a reference the percentage of serious outcome found in each cohort [34]. Our study included 275 neonates, so the results should be interpreted with caution.

One of the strengths of our study is that, being single-centred, diagnostic and therapeutic criteria are homogenous. Possible confounding factors (gestational age, female gender, percentage of intrauterine growth retardation, reason for prematurity, Apgar score <5 at 5 min of life and postnatal hydrocortisone administration) were controlled by multivariate analysis. Both cohorts showed homogeneous results in most of variables analysed and no adverse results were shown in the AM group. This is the only study with monitoring at 24 months corrected age we could find in the literature consulted.

## 5. Conclusions

The administration of antenatal betamethasone 12 mg 12 h in advance, has similar serious outcomes to the standard maturation regimen every 24 h in the preterm infant population analysed. In addition, the advance maturation shows lower hypotension during the first week of life, lower maximum FiO_2_ during admission and shorter mechanical ventilation time in the analysed population. No deleterious effect on neurodevelopment at 24 months corrected age was found in the AM group. This regimen may be an alternative in situations of imminent preterm labour where there is no time to administer the second dose within 24 h of the first dose. Nevertheless, randomised clinical trials and long-term neurodevelopmental monitoring are needed to confirm these results.

## Figures and Tables

**Figure 1 jcm-11-01227-f001:**
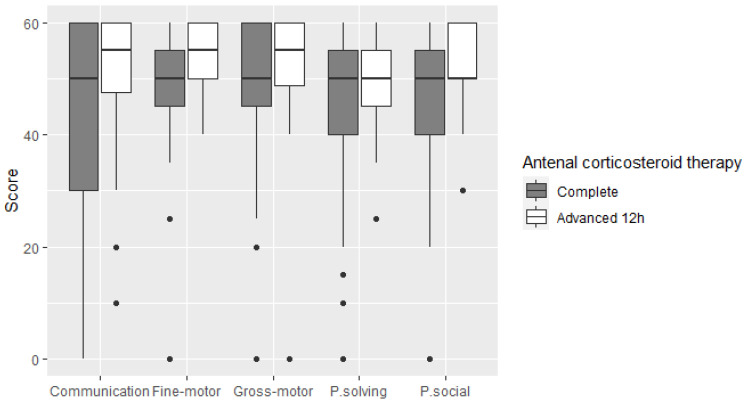
Distribution of the cohorts with respect to the ASQ^®^-3 scale. P.solving, problem solving; P.social, Personal social.

**Table 1 jcm-11-01227-t001:** Cohort characteristics with regard to the interval between antenatal corticosteroid doses.

	CM *N* = 224	AM *N* = 51	Significance (*p*) *
Maternal age (years)	32.86 ± 6.22	31.80 ± 5.46	0.23
Multiple gestation	49 (21.9%)	17 (33.3%)	0.08
Gestational diabetes	6 (3.7%)	4 (9.3%)	0.11
Hypertensive states	92 (41.4%)	12 (23.5%)	0.03
Chorioamnionitis	45 (20.1%)	15 (29.4%)	0.14
Premature rupture of membranes	68 (30.4%)	16 (31.4%)	0.88
Third trimester haemorrhage	27 (12%)	3 (5.9%)	0.21
IUGR	77 (34.4%)	10 (19.6%)	0.04
Caesarean section	184 (82.2%)	39 (76.5%)	0.75
Gestational age at first dose	27.9 ± 0.19	28.5 ± 0.38	0.14
Difference in hours from the first dose to birth	89.5 (59–122.7)	20 (16–30)	<0.001
Difference in hours from the last dose to birth	65 (35–98.7)	4 (2–12)	<0.001
Gestational age (weeks)	28.5 ± 0.2	28.7 ± 0.3	0.53
Weight at birth (grams)	991.4 ± 20.9	1087.2 ± 42.1	0.04
Length at birth (cm)	35.5 ± 0.3	36.8 ± 0.6	0.05
Head circumference at birth	25.4 ± 0.2	25.8 ± 0.3	0.39
Female gender	104 (46.4%)	31 (60.8%)	0.06
Postnatal hydrocortisone	22 (9.8%)	8 (15.7%)	0.22

Results for continuous variables are expressed as mean ± standard deviation, and as median and interquartile range. Qualitative variables show total number and percentage relative to their cohort. * Chi–square, Student’s *t*-test, Mann–Whitney U. CM: complete maturation; AM: advance maturation. IUGR: Intrauterine growth restriction.

**Table 2 jcm-11-01227-t002:** Results of the cohort comparison by prenatal corticosteroid inter-dose interval.

	CM *N* = 224	AM *N* = 51	OR (CI 95%) Unadjusted	*p* *	OR (CI 95%) Adjusted	Adjusted *p* ^†^
Serious outcome	89 (39.7%)	17 (33.3%)	0.7 (0.4–1.4)	0.39		
Neonatal death	31 (13.8%)	4 (7.8%)	0.5 (0.17–1.57)	0.24	0.5 (0.1–1.8)	0.36
Intubation at birth	125 (55.8%)	27 (52.9%)	0.9 (0.4–1.6)	0.71		
Apgar < 5 at 5 min	16 (7.1%)	2 (3.9%)	0.5 (0.1–2.3)	0.40		
Severe IVH	19 (8.5%)	4 (7.8%)	0.9 (0.3–2.8)	0.88		
PVL	8 (3.6%)	1 (2%)	0.5 (0.06–4.4)	0.56		
NEC > 2	5 (2.2%)	0	0.8 (0.7–0.8)	0.28		
Intestinal perforation	13 (5.8%)	0	0.8 (0.7–0.8)	0.13	NA	NA
Treated ROP	30 (13.4%)	6 (11.8%)	0.8 (0.3–2.1)	0.75		
Hypotension 1 week	60 (26.8%)	5 (10%)	0.3 (0.1–0.8)	0.01	0.5 (0.3–0.9)	0.04
Treated PDA	20 (8.9%)	7 (13.7%)	1.6 (0.6–4.1)	0.29		
Early sepsis	13 (5.8%)	6 (12%)	2.2 (0.7–6.1)	0.11	2.3 (0.7–7.2)	0.13
Late sepsis	99 (44.2%)	17 (34%)	0.6 (0.3–1.2)	0.18	0.8 (0.4–1.7)	0.63
BPD	50 (25.1%)	8 (17%)	0.6 (0.2–1.3)	0.23	0.6 (0.2–1.8)	0.44
Need for surfactant	122 (54.5%)	28 (54.9%)	1.01 (0.5–1.8)	0.95		
Need for MV	103 (46%)	21 (41.2%)	0.8 (0.4–1.5)	0.53		
Maximum FiO_2_	42.6 ± 2.1	34.8 ± 3.4		0.05		0.14
MV time (hours)	361.7 ± 58.5	191.1 ± 47.1		0.02		0.14

Qualitative variables are expressed as n (%) within each cohort, quantitative variables as mean ± standard deviation. * Chi–square, Fisher’s exact test, Student’s *t*-test. † Results of multivariate analyses are shown for significant variables in the unadjusted model.CM: complete maturation; AM: advanced maturation. IVH: intraventricular haemorrhage; PVL: periventricular leukomalacia; NEC: necrotising enterocolitis; ROP: retinopathy of prematurity; PDA: patent ductus arteriosus; BPD: bronchopulmonary dysplasia; MV: mechanical ventilation. NA: not applicable.

**Table 3 jcm-11-01227-t003:** Outcomes in <28 weeks gestational age according to interval between doses of antenatal corticosteroids.

	CM *N* = 94	AM *N* = 19	OR (CI 95%) Unadjusted	*p* *	OR (CI 95%) Adjusted	*p* ^†^ Adjusted
Serious outcome	70 (74.5%)	13 (68.4%)	0.7 (0.2–2.1)	0.58		
Neonatal death	26 (27.7%)	3 (15.8%)	0.4 (0.1–1.8)	0.32		
Intubation at birth	86 (91.5%)	18 (94.7%)	1.6 (0.1–14.2)	0.99		
Apgar < 5 at 5 min	11 (11.7%)	1 (5.3%)	0.4 (0.05–3.4)	0.41		
Severe IVH	16 (17%)	3 (15.8%)	0.9 (0.2–3.5)	0.89		
PVL	7 (7.4%)	0	0.6 (0.3–1.3)	0.59		
NEC > 2	-	-	-	-		
Intestinal perforation	11 (11.7%)	0	0.4 (0.2–0.8)	0.01	NA	
Treated ROP	27 (28.7%)	4 (21.1%)	0.6 (0.2–2.1)	0.49		
Hypotension 1 week	49 (52.1%)	3 (16.7%)	0.1 (0.05–0.6)	0.009	0.2 (0.05–1)	0.06
Treated PDA	18 (19.1%)	6 (31.6%)	1.9 (0.6–5.8)	0.22		
Early sepsis	11 (11.7%)	2 (11.1%)	0.9 (0.1–4.6)	0.99		
Late sepsis	61 (64.9%)	11 (61.1%)	0.8 (0.3–2.4)	0.75		
BPD	41 (56.2%)	7 (43.8%)	0.6 (0.2–1.8)	0.36		
Need for surfactant	85 (90.4%)	17 (89.5%)	0.9 (0.1–4.5)	0.89		
Need for MV	68 (72.3%)	12 (63.2%)	0.6 (0.2–1.8)	0.42		
Maximum FiO_2_	62.2 ± 3.4	44.4 ± 7.2		0.03		0.04
MV time (hours)	477.8 ± 82.4	203.1 ± 51.6		0.006		0.14

Qualitative variables are expressed as n (%) within each cohort, quantitative variables such as mean ± standard deviation. * Chi–square, Fisher’s exact test, Student’s *t*-test. ^†^ Results of multivariate analyses are shown for significant variables in the unadjusted model. vIVH: intraventricular haemorrhage; PVL: periventricular leukomalacia; NEC: necrotising enterocolitis; ROP: retinopathy of prematurity; PDA: patent ductus arteriosus; BPD: bronchopulmonary dysplasia; MV: mechanical ventilation. NA: not applicable. CM: complete maturation; AM: advanced maturation.

**Table 4 jcm-11-01227-t004:** 24 months corrected age monitoring results, according to antenatal corticosteroid regimen.

	CM *N* = 101	AM *N* = 22	OR (CI 95%)	*p* *	OR (CI 95%) Adjusted	*p*^†^ Adjusted
Cerebral palsy	3 (3%)	1 (4.5%)	1.5 (0.1–15.7)	0.71		
Hearing impairment	6 (6%)	3 (14.3%)	2.6 (0.6–11.4)	0.18	3.2 (0.7–14.9)	0.13
Visual impairment	11 (10.9%)	1 (4.5%)	2.4 (0.3–19.9)	0.68		
Score below LIN						
ASQ-communication	19 (20.2%)	2 (10%)	0.4 (0.9–2.1)	0.35		
ASQ-gross-motor	10 (10.6%9	1 (5%)	0.4 (0.05–3.6)	0.68		
ASQ-fine-motor	8 (8.5%)	-	0.5 (0.2–1.1)	0.34		
ASQ-solving	9 (9.7%)	1 (5%)	0.4 (0.05–4.11)	0.68		
ASQ-social	12 (12.8)	1 (5%)	0.3 (0.04–2.9)	0.45		
Average score						
ASQ-communication	43.6 ± 1.7	48.7 ± 3.2		0.17		0.33
ASQ-gross-motor	50.1 ± 1.1	51 ± 3.04		0.77		0.95
ASQ-fine-motor	49.8 ± 1.04	53.5 ± 1.2		0.03		0.21
ASQ-P. solving	45.05 ± 1.2	48 ± 1.8		0.20		0.45
ASQ-P. social	46.2 ± 1.2	52 ± 1.7		0.01		0.06

Qualitative variables are expressed as n (%) within each cohort, quantitative variables as mean ± standard deviation. * Chi–square, Fisher’s exact test, Student’s *t*-test. ^†^ Results of multivariate analyses are shown for significant variables in the unadjusted model. LIN: lower limit of normality. CM: complete maturation; AM: advanced maturation.

## Data Availability

The data presented in this study are available on request from the corresponding author. The data are not publicly available due to data protection policies.

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
