# Peer review of "Antenatal Betamethasone Every 12 Hours in Imminent Preterm Labour"

_jcm, 2022, doi:10.3390/jcm11051227_

Round 1

Reviewer 1 Report

Antenatal steroids play a crucial role in the management of preterm deliveries. Despite the improvement in survival and neurodevelopmental outcome in very low birth weight infants, a lot of issues still remain.

The main objective of presented study was to determine neonatal serious outcomes in preterm infants who received two doses of betamethasone 12 hours apart compared to preterm infants who received the full standard regimen.

Methods:

Two cohorts were retrospectively analyzed with regard to the type of prenatal corticosteroid administration -  complete maturation cohort (CM) iand the advanced maturation (AM) cohort. 

However, it is not clear, how these cohorts were established.

Authors should explain in details, how the imminent delivery was defined. This is essential issue in the study.

Result

Authors should explain or discuss the effect of antenatal steroids on blood pressure in infants (less hypotensive infants in advanced group).

Significant difference was also recorded in maternal hypertension. Did you treat the mothers using antihypotensive drug? If yes, which one. and how could these drugs influence the infant´s circulation. If not, do you see any other association of maternal hypertension and blood pressure in infants. 

Authors should explain, whether the mean time from the last dose to
birth (17 hours) was justified in using the AM protocol?

Conclusion

Auhors should be careful in recommendation of advanced maturation. Please adjust accordingly. 

Author Response

Dear Reviewer
First of all, thank you for your critical reading of the article and your considerations to improve it.
We have added your suggestion about the definition of imminent preterm birth, as well as the indications by the obstetrics service to administer the second dose of betamethasone 12 hours after the first.
Regarding hypotension, we have added the hypotensive treatment received by the mothers, as you suggested, it can influence the newborn's blood pressure. In our analysis, the differences in favor of the 12-hour group persist even though this confounding factor has been included in the multivariate analysis.
Likewise, we have added a secondary analysis, excluding those premature infants with mothers affected by preeclampsia, eclampsia or HELLP syndrome, finding that the 12-hour group shows less hypotension with p = 0.007.
Thank you very much for your comments about the time difference between administration of the second dose and delivery in the 12-hour cohort. When exposing our results as mean and standard deviation, they have been altered by extreme values. We have considered it prudent to change them to median and interquartile range for better definition.
Regarding the conclusions, we recommend more studies on this topic, since our results are with our limited population.

Thank you very much for your comments
We hope that the modifications are to your liking.

Reviewer 2 Report

The authors reported a retrospective cohort study including preterm infants ≤34 weeks gestational age and birth and ≤1500g admitted in a NICU IIIC level in a Tertiary Hospital from 2015 to 2020 to determine whether the administration of two doses of betamethasone in a 12 hours interval is equivalent to the effects of a full maturation.

The manuscript is very well-written. The objective, methods, primary outcome, statistical analyses are very well-defined and explained.

Some minor modifications should be done:

  • In abstract: line 26, the authors should modify as “Neurodevelopment at 2 years was evaluated with the validated Ages & Stages Questionnaires®, Third Edition (ASQ®-3)”.
  • In abstract, the authors should spell “MV (line 31)
  • In methods, line 84, “threatening conditions” should be defined
  • In methods, line 140, the date of approval should be added
  • In discussion, lines 263-265, the authors should explain how the sample size calculation was done and what references were used.

Author Response

Dear Reviewer
First of all, thank you for your critical reading of the article and your considerations to improve it.
We have modified lines 26 and 31 according to your suggestions.
As for line 84, “threatening conditions” is a translation error, since it referred to major or life-threatening malformations. We have suppressed it since we have not included any patient with malformations.
We have added the date of approval by the provincial ethics committee of Malaga in line 140.
We are very grateful for your point in lines 263-265 about sample size. It has allowed us to correct a calculation error, finding that the sample size necessary to detect a 5% difference in serious outcome, according to its frequency in our cohorts, with a confidence interval of 95%, is 808 patients and not 58 as we had reflected in the beginning.

We hope that these modifications are to your liking.
Thank you very much for your evaluation.